# Effect of Palivizumab Prophylaxis on Respiratory Syncytial Virus Infection in Very Preterm Infants in the First Year of Life in The Netherlands

**DOI:** 10.3390/vaccines11121807

**Published:** 2023-12-02

**Authors:** Rutger M. Schepp, Joanna Kaczorowska, Pieter G. M. van Gageldonk, Elsbeth D. M. Rouers, Elisabeth A. M. Sanders, Patricia C. J. Bruijning-Verhagen, Guy A. M. Berbers

**Affiliations:** 1Centre for Infectious Disease Control, National Institute for Public Health and the Environment, P.O. Box 1, 3720 BA Bilthoven, The Netherlands; joanna.kaczorowska@rivm.nl (J.K.);; 2Julius Centre for Health Sciences and Primary Care, University Medical Centre Utrecht, 3508 GA Utrecht, The Netherlands; 3Department of Paediatric Immunology and Infectious Diseases, Wilhelmina Children’s Hospital, 3508 AB Utrecht, The Netherlands

**Keywords:** RSV prophylaxis, palivizumab effectiveness, very preterm infants, RSV infection rates, gestational age

## Abstract

Respiratory Syncytial Virus (RSV) poses a severe threat to infants, particularly preterm infants. Palivizumab, the standard preventive prophylaxis, is primarily utilized in high-risk newborns due to its cost. This study assessed palivizumab’s effectiveness in preventing RSV infections in predominantly very preterm infants during their first year of life. Serum samples from a prospective multicentre cohort study in the Netherlands were analyzed to assess RSV infection rates by measuring IgG levels against three RSV proteins: nucleoprotein, pre-fusion, and post-fusion protein. Infants were stratified based on gestational age (GA), distinguishing very preterm (≤32 weeks GA) from moderate/late preterm (>32 to ≤36 weeks GA). In very preterm infants, palivizumab prophylaxis significantly reduced infection rates (18.9% vs. 48.3% in the prophylaxis vs. non-prophylaxis group. Accounting for GA, sex, birth season, and birth weight, the prophylaxis group showed significantly lower infection odds. In infants with >32 to ≤36 weeks GA, the non-prophylaxis group (55.4%) showed infection rates similar to the non-prophylaxis ≤32-week GA group, despite higher maternal antibody levels in the moderate/late preterm infants. In conclusion, palivizumab prophylaxis significantly reduces RSV infection rates in very premature infants. Future research should explore clinical implications and reasons for non-compliance, and compare palivizumab with emerging prophylactics like nirsevimab aiming to optimize RSV prophylaxis and improve preterm infant outcomes.

## 1. Introduction

The Respiratory Syncytial Virus (RSV) is highly contagious and commonly infects children before the age of 2 years [1,2]. Severe RSV infections, particularly among infants under 6 months old, can lead to serious lower respiratory tract infections such as bronchiolitis and pneumonia, resulting in significant hospitalization rates and mortality in young children [3,4]. In low- and middle-income countries, RSV infections cause approximately 101,400 deaths in children under five annually, with the highest rates in those aged 28 days to 6 months [5]. This burden is not limited to low-income countries, as a high incidence of RSV-associated hospitalizations was observed also in high-income European countries. Remarkably, one in every fifty-six healthy term-born infants in these countries is affected, with the highest risk observed in infants younger than 3 months. Such infections necessitate prolonged hospital stays and are associated with a higher likelihood of intensive care unit admissions compared to non-RSV lower respiratory tract infections [6].

While the majority of RSV hospitalizations are among otherwise healthy children, premature infants born before 37 weeks of gestational age face an even higher risk of complications, long-term morbidities, and hospitalization due to physiological, metabolic, and immunological differences, irrespective of underlying diseases [7,8,9]. Furthermore, maternal transfer of IgG antibodies reaches the highest levels in the final trimester [10,11], impacting protection against RSV infection in the first 6 months of life.

Until recently, the only available intervention to protect newborns in the first months of life against severe RSV infection was passive immunization with palivizumab (Synagis), a humanized IgG1ĸ monoclonal antibody directed at the RSV fusion (F) protein. The RSV F protein plays a crucial role in viral infection by mediating fusion between the viral and targeting cell membranes through a conformational change. Of particular importance is the pre-fusion (Pre-F) form of the fusion protein, which initiates fusion in the cellular membrane and elicits more efficient and potent neutralizing antibodies compared to the post-fusion (Post-F) form [12]. Palivizumab targets an epitope present on both Pre-F and Post-F forms of the fusion protein [13,14]. Administered once a month at a dose of 15 mg/kg intramuscularly, palivizumab has a half-life of 17–27 days. However, its high costs and uncertainty regarding cost-effectiveness have limited its use to high-risk groups, such as infants with prematurity ≤32 weeks of gestation and/or bronchopulmonary dysplasia [15,16]. Palivizumab prophylaxis has demonstrated significant reductions (up to 80%) in hospitalizations, depending on the presence of co-morbidities and/or the degree of prematurity [14,17,18,19]. Premature infants, particularly those born at ≤32 weeks of gestational age, face a higher risk of complications, including respiratory distress syndrome, bronchopulmonary dysplasia, and intraventricular hemorrhage [8,20,21]. To address these risks, effective postnatal prevention of RSV infection is crucial. In the Netherlands, prophylaxis with the monoclonal antibody palivizumab is offered to preterm infants with a gestational age ≤32 weeks who are under six months old at the beginning of the RSV season, in accordance with healthcare insurance provisions [22,23]. Typically, palivizumab administration aligns with the discharge of infants from the neonatal intensive care unit, usually occurring at around 1 to 2 months of age.

The primary objective of this study was to assess the effectiveness of palivizumab prophylaxis in preventing RSV-related infections among very preterm infants during their first year of life. The presence of natural RSV infection was determined through measurements of IgG levels specific to three RSV proteins: nucleoprotein (N), Post-F, and Pre-F. We compared the infection rates between premature infants who received palivizumab prophylaxis and those who did not, with assessments conducted at approximately 5 and 12 months post-birth. 

This study provides valuable insights into the prevalence of RSV infection among very preterm infants during their first year of life and assesses the effectiveness of palivizumab prophylaxis within this vulnerable demographic. It is especially noteworthy since studies concerning a recently approved alternative to palivizumab, nirsevimab, in very/extremely premature infants are currently lacking. The reduction in RSV infections not only logically forbodes decreased hospitalization but also holds promise for substantial health benefits, especially given that mild RSV infections are implicated in recurrent wheezing and asthma, even among healthy (pre)term infants [24].

## 2. Methods

### 2.1. Study Design and Participants

This study utilized serum samples from a large prospective, observational, multicenter, cohort study conducted in the Netherlands, involving preterm infants during their first year of life. Infants were recruited from 8 hospitals across the Netherlands between October 2015 and October 2017 with a follow-up until 12 months of age (November 2018) [25]. Written informed consent was obtained from both parents or guardians of the infants before enrolment. This study received ethical approval from the Medical Ethical Committee of the University Medical Centre Utrecht and adhered to Good Clinical Practice guidelines and the Declaration of Helsinki. It was registered with the Netherlands Trial Register (NL7142) [26]. The cohort study followed standard practices of the Dutch immunization program for administering vaccinations and RSV prophylaxis. Capillary blood samples were collected at three time points: before the start of the primary vaccinations (approximately 6 weeks after birth), one month after the last dose of the primary series (at approximately 5 months of age), and one month after the booster vaccination (at approximately 12 months of age) (Figure 1). 

A total of 276 preterm infants, with a gestational age (GA) of ≤36 weeks, participated in the study. Seven infants were excluded due to a lack of remaining serum samples (Figure 1). For this analysis, the remaining 269 infants were categorized into two distinct groups based on gestational age: those born at ≤32 weeks and those born at >32 to ≤36 weeks. This stratification allowed for the examination of the effect of palivizumab prophylaxis on RSV infection while accounting for the physiological and clinical differences between these two populations. Very premature infants born at ≤32 weeks GA face significantly higher risks when compared to their moderate/late preterm (GA > 32 to ≤36 weeks) counterparts. Furthermore, in the Netherlands, palivizumab prophylaxis is only indicated for infants with a GA of less than 32 weeks. It is prescribed for those with a GA greater than 32 weeks solely when they have severe cardiovascular comorbidities or are afflicted by bronchopulmonary dysplasia. Within both GA groups, we identified three palivizumab prophylaxis categories: “Palivizumab (Pal+)”, “No Palivizumab (Pal−)”, or “Unknown”. 

### 2.2. Assay

The assay involved the concurrent examination of serum samples for the presence of immunoglobulin G (IgG) antibodies targeting the RSV prefusion F protein (Pre-F), postfusion F protein (Post-F), and nucleoprotein (N) through an RSV multiplex immunoassay (MIA) [27]. In brief, each of the three RSV antigens was coupled to distinct color-coded activated carboxylated beads. Serum samples were diluted 1/200 and 1/8000 in a solution containing phosphate-buffered saline with 1% bovine serum albumin and 0.1% Tween 20. These diluted samples were then incubated with the conjugated beads, followed by incubation with R-phycoerythrin-labeled goat anti-human IgG antibody (Jackson Immunoresearch). Data acquisition for the samples was executed utilizing a Flexmap 3D (Luminex Corp.) in conjunction with xPONENT 4.3 software (DiaSorin/Luminex). The quantification of serum IgG antibody concentrations was carried out in arbitrary units/mL (AU/mL) by interpolation from a five-parameter logistic curve derived from an in-house reference serum, using Bio-Plex Manager software version 6.2 (Bio-Rad Laboratories, Hercules, CA, USA).

### 2.3. Outcomes

The primary outcome of this study was to assess the impact of palivizumab prophylaxis on the rate of RSV infection in preterm infants during the first year of life. Infection status was determined at the 12-month mark by measuring the concentrations of Pre-F and N-specific IgGs. It is important to note that palivizumab, as a humanized IgG antibody, binds to Post-F and Pre-F, which interferes with the accurate quantification of endogenous Post-F and Pre-F IgG antibodies. To address this, we exclusively examined antibody levels specific to the RSV N protein in infants who received palivizumab. Conversely, in infants without palivizumab prophylaxis, we assessed both Pre-F and N antibodies to detect infection. 

Interpretation of infection status could be influenced by the presence of maternal IgGs. Maternal IgG levels wane completely within 10–12 months, with an estimated half-life of approximately 42 days [1], consequently resulting in roughly 10- and 100-fold lower IgG concentration at 5 and 12 months, respectively, compared to those at 6 weeks. As RSV antibodies elicited by natural infection wane at a slower rate [28], we defined a primary RSV infection based on two criteria: (1) IgG levels equaled or exceeded previously established cut-offs for at least one of the two IgGs (2 AU/mL for Pre-F and 8 AU/mL for N; [29]) and (2) the concentration equaled or exceeded the previous time point’s level. Both criteria were required to classify an infection.

Secondary outcomes included assessing the infection rate in infants in the GA group > 32 to ≤36 weeks at 12 months. Additionally, we examined maternal IgG levels before any RSV prophylaxis was administered (at 6 weeks) to evaluate the rate of maternal IgG transfer across different gestational ages. For this analysis, we included all premature infants with antibody status determined at 6 weeks, including those >32 to ≤36 weeks and irrespective of palivizumab prophylaxis. We also compared IgG antibody trajectories over 3 time points (6 weeks, 5 months, and 12 months) for RSV proteins (N, Pre-F, and Post-F) in infected and uninfected children within the GA ≤ 32 weeks subgroup. 

### 2.4. Statistical Analysis

We conducted statistical analyses using R version 4.3.0 (stats package) and GraphPad Prism (version 9.5.1) for figure construction. To compare the differences in geometric mean antibody concentrations between groups, we employed the Wilcoxon sum-rank test. We calculated the proportions of infants with natural infection and those without infection separately for two groups: those who received palivizumab and those who did not. The Chi-squared test with Yates’s continuity correction was used to compare proportions at each sampling point.

To evaluate the effect of palivizumab prophylaxis on the odds of experiencing RSV infection, we employed a binomial multivariate logistic regression model. The model included the following predictors: sex, birthweight, gestational age, birth season, and presence of palivizumab prophylaxis, without interactions. The model was constructed using the stats package and visualized using the sjPlot package (v. 2.8.14) in R. 

Bonferroni-adjusted *p* values < 0.05 were considered statistically significant. 

## 3. Results

### 3.1. Baseline Characteristics

A total of 269 infants were included in the study, consisting of 152 males and 117 females (Figure 1, Table 1). The infants were classified into two gestational age (GA) groups: GA ≤ 32 weeks (n = 188) and GA > 32 to ≤36 weeks (n = 81). In the GA ≤ 32 weeks group, 135 infants (66.2%) received palivizumab at least once, while 31 infants (21.7%) did not. Among the GA > 32 to ≤36 weeks group, the majority of premature infants did not receive palivizumab (72 infants, 88.9%), except for two infants with a GA of 32 weeks and 1 day who received the prophylaxis. Additionally, twenty-nine infants (22 in the GA ≤ 32 weeks group and seven in the GA > 32 to ≤36 weeks group) had an unknown RSV prophylaxis status and were excluded from further analysis.

In the GA ≤ 32 weeks group, the infants who received palivizumab (n = 135) had a mean GA of 28.3 weeks (range: 24.0–32.0) and a mean birth weight of 1110.0 g (range: 530–2142). Infants who did not receive palivizumab (n = 31) had a mean GA of 29.4 weeks (range: 25.3–32.0). The mean birth weight was significantly higher than in the infants who received prophylaxis, with a mean of 1338.0 g (range: 595–2034; Wilcoxon test, *p* = 0.0032). The proportion of infants receiving palivizumab was significantly higher among those born in the spring/summer months compared to the fall/winter months (Chi-squared test, *p* = 0.0014; Table 1). Conversely, the distribution of infants who did not receive Palivizumab was approximately equal between the fall/winter and spring/summer months of 2015 to 2017 (*p* = 0.21). 

In the GA > 32 to ≤36 weeks group, infants who did not receive palivizumab (n = 72) had a mean GA of 34.0 weeks (range: 32.1–36.0) and a mean birth weight of 2048.6 g (range: 1205–3355) (Table 1). Notably, the only two infants (male twins) within this group who were administered prophylaxis had a GA of 32.1 weeks and birth weights of 1680 g and 1715 g, respectively. Due to the limited number of infants receiving palivizumab in this GA group, these two were excluded from further analysis. 

### 3.2. Antibody Concentrations 

The levels of nucleoprotein-specific serum IgG (N IgG) and pre-fusion protein-specific serum IgG (Pre-F IgG) were measured at three time points (6 weeks, 5 months, and 12 months after birth) in both GA groups. Among infants with GA ≤ 32 weeks, in the palivizumab prophylaxis (Pal+) subgroup and the non-prophylaxis subgroup (Pal−), the N IgG concentrations were similar at 6 weeks (Pal+: 32.0 AU/mL [95% CI: 28.0–40.2]; Pal−: 36.5 AU/mL [95% CI: 25.9–51.4], *p* = 0.5477) and at 5 months (Pal+: 3.62 AU/mL [95% CI: 2.94–4.46]; Pal−: 4.51 AU/mL [95% CI: 2.82–7.21], *p* = 03103), but significantly lower at 12 months in the Pal+ subgroup (Pal+: 1.82 AU/mL [95% CI: 1.2–2.76]; Pal−: 13.1 AU/mL [95% CI: 3.59–47.9]; *p* = 0.0031) (Figure 2A: right panel). The Pal+ and Pal− subgroups exhibited comparable Pre-F antibody geometric mean concentrations (GMC) at 6 weeks (Pal+: GMC 33.5 AU/mL [95% CI: 28.0–40.2]; Pal−: GMC 47.5 AU/mL [95% CI: 32.0–70.7]; Wilcoxon test, *p* = 0.0591). However, the Pal+ subgroup had significantly higher Pre-F IgG concentrations at 5 months (Pal+: 633 AU/mL [95% CI: 418–958]; Pal−: 9.1 AU/mL [95% CI: 4.4–19.1]; *p* < 0.0001) and at 12 months (Pal+: 267 AU/mL [95% CI: 184–388]; Pal−: 2.88 AU/mL [95% CI: 0.86–9.59]; *p* < 0.0001) due to palivizumab interference (Figure 2A: left panel). 

At 6 weeks, significantly lower maternal antibody concentrations were observed in the very preterm infants (GA ≤ 32 weeks) compared to the moderate/late preterm infants (GA > 32 to ≤36 weeks) for both Pre-F IgG (*p* = 0.0017) and N IgG (*p* < 0.0001) (Figure 2B) in line with lower rates of maternal IgG transfer at younger GA. The results for Post-F were consistent with those of Pre-F (Appendix A). In the GA group > 32 to ≤36 weeks, the GMCs for Pre-F IgG were 84.3 AU/mL [95% CI 70.4–101], 9.01 AU/mL [95% CI: 6.95–11.7], and 5.48 AU/mL [95% CI: 2.64–11.4] at 6 weeks, 5 months, and 12 months, respectively. As for N protein IgG, the GMCs were 113 AU/mL [95% CI 93.1–137], 16.3 AU/mL [95% CI: 11.6–23.1], and 14.3 AU/mL [95% CI: 7.39–27.5] at 6 weeks, 5 months, and 12 months, respectively (Figure 2B). 

### 3.3. Incidence of RSV Infection 

The primary objective of this study was to compare RSV infection rates in very preterm infants receiving palivizumab prophylaxis with those who did not, during their initial 12 months of life. In the group with GA ≤ 32 weeks, 18.9% of prophylaxis-receiving infants exhibited indications of natural RSV infection within the first year of life (Table 2). In contrast, the non-prophylaxis group had a significantly higher percentage of RSV infections with 48.3% of infants affected (Chi-squared test, *p* = 0.00203). 

In the subgroup of infants with a GA >32 to ≤36 weeks, we observed a 55.4% infection rate in the non-prophylaxis group. Within this GA group, only two infants were palivizumab recipients (GA of 32 weeks and 1 day), and both contracted an RSV infection.

To address potential confounders when quantifying the effectiveness of the palivizumab prophylaxis within the GA ≤ 32 group, we constructed a binomial logistic multivariate model, which considered the presence of infection as a response variable and included, besides the presence of prophylaxis, the following variables: sex, birthweight, gestational age, and birth season. Pal+ infants had significantly lower odds of getting infected (odds ratio = 0.25 [95%CI = 0.099–0.621], *p*-value = 0.0029) than Pal− infants. There was no significant effect of sex, birth season, gestational age, and birth weight on infection status (Appendix A).

### 3.4. Antibody Level Trajectories

To gain insights into antibody level trajectories in infected and non-infected children, we examined IgG concentrations across three time points post-birth: 6 weeks, 5 months, and 12 months. We analyzed three groups: palivizumab-receiving infants with GA ≤ 32 (Figure 3), non-palivizumab infants with GA ≤ 32 (Appendix A), and non-palivizumab infants with GA > 32 (Appendix A).

As anticipated, nucleoprotein levels gradually declined in uninfected infants across all three subgroups as maternal antibodies diminished over time (Figure 3A, Appendix A). We observed a similar pattern in Pre-F and Post-F specific IgG levels in uninfected children without prophylaxis, both in the GA ≤ 32 and GA > 32 to ≤36 weeks groups (Appendix A).

Among palivizumab-receiving children, Pre-F (Figure 3B) and Post-F (Figure 3C) levels increased compared to the previous time point at either 5 months (78 children, 66 uninfected), 12 months (16 children, 12 uninfected), or both time points (11 children, 9 uninfected). One uninfected palivizumab recipient exhibited no increase in Pre-F or Post-F levels compared to the previous time point but had a notably high Pre-F concentration (1752 Au/mL) at the 6-week mark, suggesting probable palivizumab administration around that period.

Given that 24 out of 135 children still experienced an infection despite prophylaxis, we conducted a post hoc analysis to compare antibody concentrations at month 5 between children who experienced an infection between the 5th and 12th month after birth and those who did not. Children who experienced an infection by 12 months tended to have significantly lower Pre-F (*p* = 0.028) and Post-F (*p* = 0.028) levels (at 5 months) than the ones with no evidence of infection. However, there were eighteen Pal+ children (four with infections at 12 months) who had lower Pre-F and Post-F concentrations at month 5 than at week 6 (likely due to prophylaxis initiation after 5 months). After excluding these 18 individuals, the Pre-F concentrations in the uninfected remained marginally significantly higher than in the infected (*p* = 0.046), while the significance was lost (*p* = 0.071) for Post-F.

## 4. Discussion

The main objective of this study was to assess the effectiveness of palivizumab prophylaxis in preventing RSV-related infections among very preterm infants (GA ≤ 32 weeks), born in the Netherlands, during their first year of life. Additionally, we investigated differences in IgG antibody levels against nucleoprotein (N) and pre-fusion (Pre-F) protein across different gestational age groups. These findings are of particular importance considering the paucity of data available for this vulnerable population of very premature infants. It is also noteworthy that existing studies on nirsevimab, an alternative prophylactic agent, have primarily focused on healthy late-preterm and term infants [30,31].

Our study revealed a significant reduction in RSV infection rates among infants receiving palivizumab prophylaxis compared to those without prophylaxis. Specifically, in infants born at ≤32 weeks of gestational age, the palivizumab prophylaxis group demonstrated a substantially lower infection rate (18.9%) compared to the non-prophylaxis group (48.3%). Similarly, for infants born at >32 to ≤36 weeks of gestational age, the non-prophylaxis group exhibited a comparable infection rate (55.4%) to the non-prophylaxis ≤32 weeks GA group. These infection rates align with recent population studies in term-born infants [2,25], emphasizing the relevance of our findings to the broader context of RSV infections.

The 2.6-fold lower incidence and 4-fold lower odds of RSV infection in very preterm infants receiving palivizumab provide robust evidence for the effectiveness of this prophylaxis, particularly during the first year of life. Furthermore, our comprehensive analysis within this very preterm group, including variables such as gestational age, birth weight, birth date, and gender, did not reveal any significant impact on infection rates. These results are consistent with previous studies reporting the efficacy of palivizumab in reducing hospitalizations and RSV-associated infections [17,32].

Despite lower maternal antibody transfer in very preterm infants, evidenced by significantly lower IgG concentrations against Pre-F and N protein at 6 weeks of age when compared to late preterm infants, infection rates were similar in both groups over the initial 12 months of life when no palivizumab was administered. Moreover, infants who experienced an infection between 5 and 12 months despite palivizumab administration showed marginally significantly lower Pre-F IgG concentration at 5 months of age (*p* = 0.046) compared to those that did not experience an infection. This observation highlights the potential importance of maintaining an optimal level of exogenous prophylactic monoclonal antibodies for effective protection throughout the first year of life.

However, our study has limitations. The original study was designed to evaluate the immunogenicity of routine vaccinations in the first year of life in preterm infants and did not focus on RSV disease and prophylaxis. This resulted in the absence of critical data on disease severity, morbidities, and hospitalization outcomes. Incomplete data on palivizumab administration frequency and specific administration hindered certain analyses related to seasonality and the efficacy of multidose palivizumab administration. Approximately 21% of very preterm infants did not receive palivizumab despite the RSV intervention guidelines. Further exploration of the reasons behind this non-compliance, such as parental decision-making or concerns about side effects, is warranted to improve adherence rates and optimize the benefits of prophylactic interventions. Birthweight may have influenced the clinicians’ choice to withhold palivizumab since the birthweight was significantly higher in the non-prophylaxis subgroup within the GA ≤ 32 weeks group. However, our multivariate analysis suggests that its impact on infection rates over the entire 12 months is likely minimal. 

The strength of this study lies in the substantial number of preterm infants, particularly in the vulnerable very preterm group (n = 188). This enabled a robust comparison of infection rates, providing new insights into palivizumab’s real-world effectiveness. Our findings add valuable data specific to this high-risk population, which are crucial given the healthcare burden of RSV infections in preterm infants. Palivizumab’s ability to lower infection rates can alleviate this healthcare burden, thereby reducing the need for costly hospitalizations and intensive medical interventions. Moreover, by preventing RSV infections and their associated complications, palivizumab has the potential to improve long-term health outcomes in this vulnerable population.

The emergence of novel prophylactic agents, such as the monoclonal antibody nirsevimab (2023), has introduced new possibilities for RSV prevention. Nirsevimab’s extended half-life and single-dose administration offer potential advantages over palivizumab, including improved compliance and reduced costs associated with multi-dose administration. Furthermore, nirsevimab demonstrated promising efficacy in both preterm and healthy term infants [30,31], making it a compelling cost-effective choice for RSV prophylaxis in late preterm infants or even healthy term infants, where palivizumab lacks cost-effectiveness [33]. 

While maternal vaccination for RSV prevention is still in the early stages of development, its effectiveness among preterm infants is anticipated to be notably reduced due to the extended vaccination window (up to 36 weeks GA) and the imperative need for a timely vaccine response and subsequent efficient transfer of RSV-specific antibodies to the developing fetus. Recent phase 3 trials have yielded mixed results, with some demonstrating limited efficacy [34,35] and others raising safety concerns [36]. 

The exclusion of women with high-risk pregnancies and those predisposed to prematurity from these trials accentuates potential limitations and associated risks of maternal RSV vaccinations. In essence, even after maternal RSV vaccination becomes available, its effectiveness among preterm infants is anticipated to be notably reduced, thereby underscoring the continued importance of RSV prophylaxis. While the influence of gestational age and birth date on RSV infection remains debatable, our analyses indicate that gestational age and birth date of preterm infants do not significantly impact RSV infection risk, as the infection rates between the non-prophylaxis age groups are comparable to those in healthy term-born infants. However, it is likely that gestational age does influence the severity of the disease [7,8,9], but this study lacks specific data on disease severity. 

## 5. Conclusions

Our study demonstrates that palivizumab prophylaxis significantly reduces the infection rate in very premature infants (<32 weeks gestational age). Despite variations in palivizumab frequency and timing, our findings clearly demonstrate its efficacy in lowering infection rates, which logically translates into a reduction in hospitalizations. The substantial reduction in RSV infections observed in the palivizumab prophylaxis group highlights the effectiveness of this intervention in preventing RSV-related morbidity in this vulnerable population. Further research should delve into the clinical implications of palivizumab prophylaxis, explore the underlying reasons for non-compliance, and compare its efficacy against emerging prophylactic options such as nirsevimab. Addressing these aspects will contribute to the optimization of RSV prophylaxis strategies, thereby enhancing outcomes for preterm infants.

## Figures and Tables

**Figure 1 vaccines-11-01807-f001:**
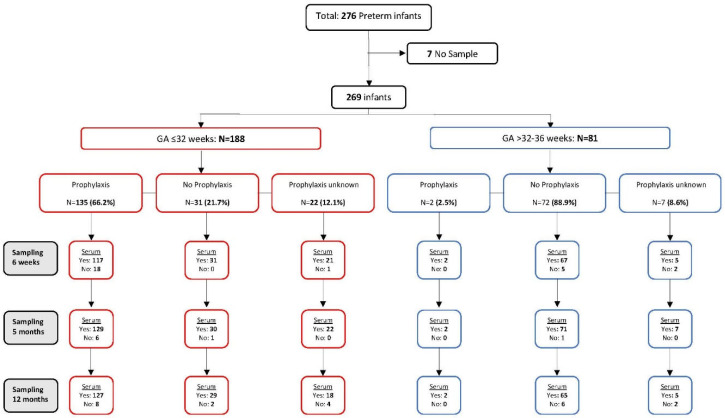
Study design and participant flowchart. Flow diagram of participants throughout the study, starting with the total number of infants initially included. It shows the stratification of participants based on gestational age (GA) and the administration of palivizumab prophylaxis.

**Figure 2 vaccines-11-01807-f002:**
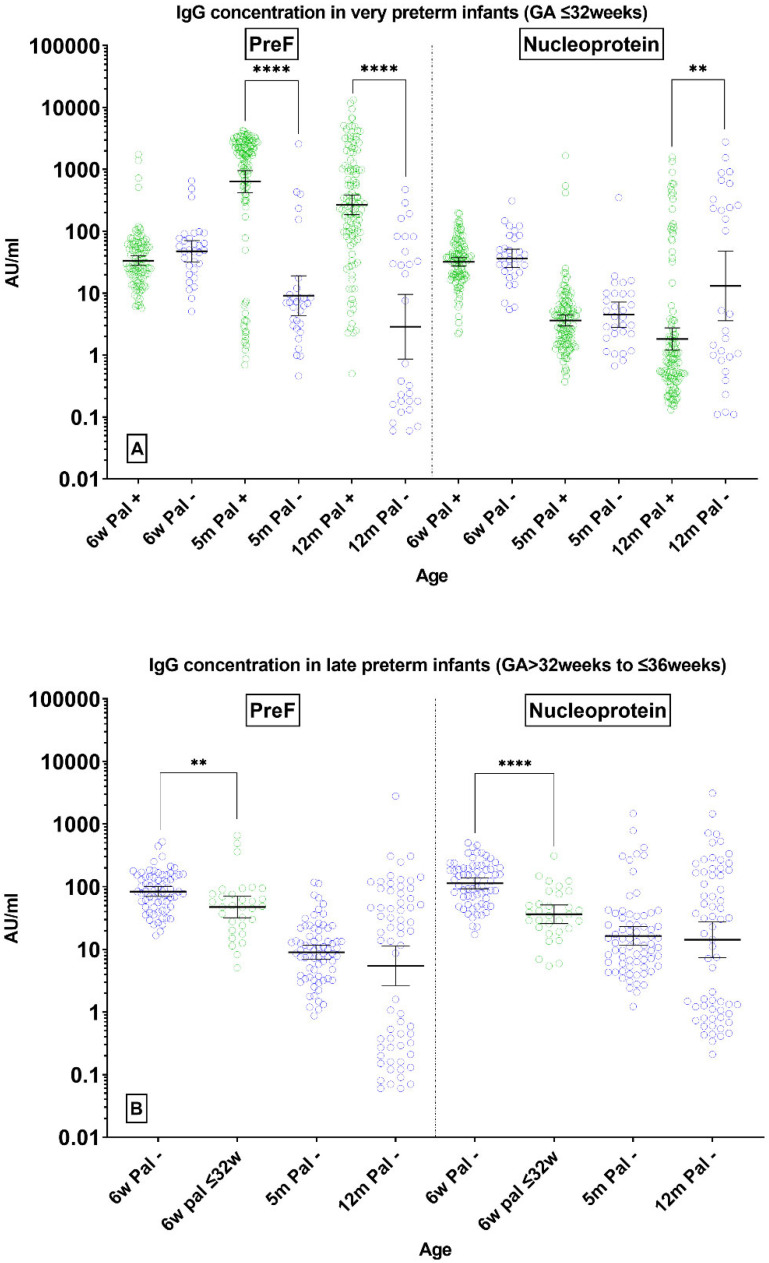
Antibody concentrations (AU/mL) in preterm infants. (**A**) Serum IgG levels specific to pre-fusion protein (Pre-F IgG) and nucleoprotein (N IgG) in infants with GA ≤32 weeks at 6 weeks, 5 months, and 12 months. Comparisons between palivizumab prophylaxis (Pal+) and non-prophylaxis (Pal−) groups were evaluated using the Wilcoxon test. (**B**) Pre-F IgG and N IgG concentrations in infants with GA > 32 to ≤36 weeks at corresponding time points. Maternal antibody transfer was examined between non-prophylaxis GA subgroups at 6 weeks using the Wilcoxon test. (** *p*-value < 0.01, **** *p*-value < 0.0001).

**Figure 3 vaccines-11-01807-f003:**
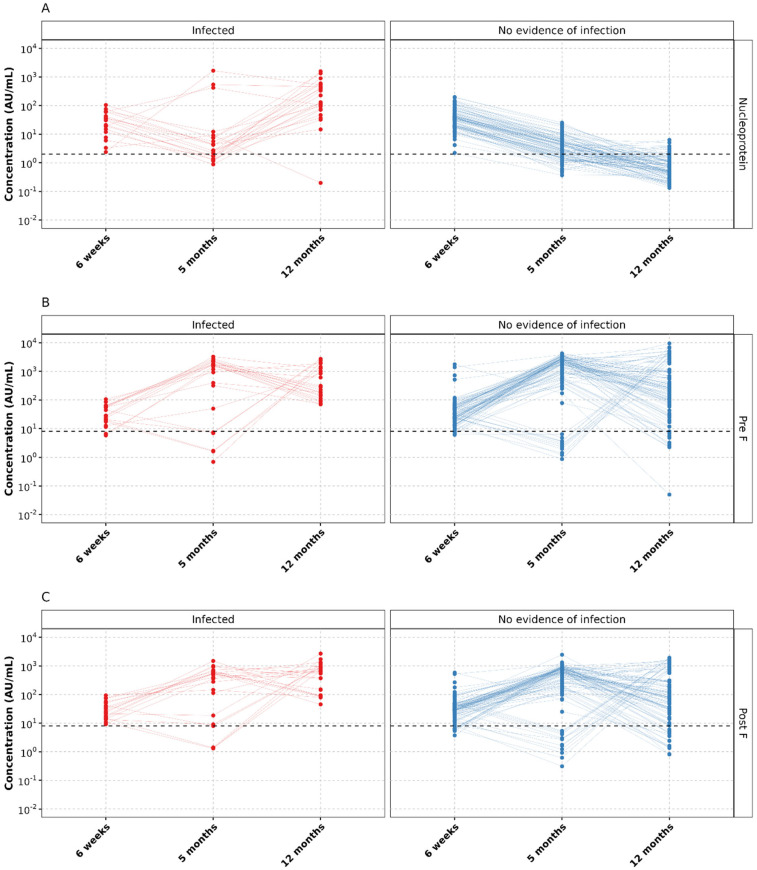
IgG concentrations in children from GA < 32 weeks subgroup who received palivizumab stratified by the infection status. IgG specific to (**A**) nucleoprotein, (**B**) Pre-F, (**C**) Post-F. The horizontal dashed line represents the cutoff of the assay.

**Table 1 vaccines-11-01807-t001:** Characteristics of infants included in the study.

	Total (n = 269)	GA ≤ 32 w PAL+ (n = 135)	GA ≤ 32 w PAL− (n = 31)	GA >32 w–≤36 w PAL+ (n = 2)	GA >32 w–≤36 w PAL− (n = 72)	Unknown (n = 29)
Sex, n (%)						
Male	152 (56.5%)	78 (57.8%)	19 (61.3%)	2 (100%)	36 [50.0%]	17 (59.6%)
Female	117 (43.5%)	57 (42.2%)	12 (38.7%)	0 (0.0%)	36 [50.0%]	12 (41.4%)
Gestational age (weeks), mean [range]	30.2 [24.0–36.0]	28.3 [24.0–32.0]	29.4 [25.3–32.0]	32.1 [32.1–32.1]	34.0 [32.1–36.0]	30.1 [26.6–35.3]
Birth weight (g), mean [range]	1427 [530–3355]	1110.0 [530–2142] **	1338.0 [595–2034] **	1697.5 [1680–1715]	2048.6 [1205–3355]	1438 [800–2775]
Born “Summer” April–September	157 (58.4%)	86 (63.7%) *	12 (38.75)	0 (0.0%)	44 (66.1%)	15 (48.3%)
Born “Winter” October–March	112 (41.6%)	49 (36.3%) *	19 (61.35)	2 (100%)	28 (38.9%)	14 (51.7%)

Data are means with range or number with proportions (%). * Significant difference *p* = 0.0014 chi-squared test, ** Significant difference *p* = 0.0032 Wilcoxon sum rank test. Note: PAL+ = received palivizumab, PAL− = did not receive palivizumab.

**Table 2 vaccines-11-01807-t002:** Infection status between groups.

Infection	With Prophylaxis GA ≤ 32 w			Without Prophylaxis GA ≤ 32 w			Without Prophylaxis GA > 32 w ≤36 w		
	Yes	No	n	Yes	No	n	Yes	No	n
≤5 Months	4 (2.3%)	125 (97.7%)	129	3 (10.0%)	27 (90.0%)	30	8 (11.3%)	63 (88.7%)	71
6–12 Months	20 (15.7%)	107 (84.3%)	127	11 (37.9%)	18 (62.1%)	29	28 (43.1%)	37 (56.9%)	65
Total at 12 Months	24 (18.9%) *	103 (81.1%)	127	14 (48.3%) *	15 (51.7%)	29	36 (55.4%)	29 (44.6%)	65

Data are number with proportions (%).* Significant difference *p* = 0.00203 chi-squared test. Note: GA—Gestational age.

## Data Availability

The data presented in this paper are available on request from the corresponding author.

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
