# Peer review of "Effect of Palivizumab Prophylaxis on Respiratory Syncytial Virus Infection in Very Preterm Infants in the First Year of Life in The Netherlands"

_vaccines, 2023, doi:10.3390/vaccines11121807_

Round 1

Reviewer 1 Report

Comments and Suggestions for Authors

The manuscript entitled “Effect of Palivizumab prophylaxis on RSV infection in very preterm infants in the first year of life in the Netherlands” by Schepp et al. describes the effect of palivizumab prophylaxis in preventing RSV infections in very preterm (<32 weeks) and preterm (>32-37 weeks) infants for their first year of life. IgG levels were measured for three RSV proteins nucleoprotein (N), pre-fusion F and post-fusion F for the evaluation of RSV infection rates. Authors have analyzed the serum for the very preterm and preterm infants born in Netherlands. Interestingly, the infection rates remain same in the non-prophylactic groups in both very preterm (<32 weeks) and preterm (>32-37 weeks) infants. Sex, birth season and birth weight were also considered in the study. Overall, this is an important study, and the data has been analyzed carefully. However, it requires few changes and need additional information for further improvement.

The manuscript looks very repetitive in abstract, introduction as well as discussion. Discussion looks more like a review paper. For the current study, this much review of literature looks unnecessary and only create confusion and additional questions. Please rewrite it for better read and be concise to the current study.

The study does not include the data if the infants were asymptomatic or mild/severe illness. Or if some data is available about the hospitalization. And the study could be followed up for one more year because of high risk of RSV infection in first 2 years.

What is the timeline for the data collection? Do authors have the information about the time-period during which these data were collected and analyzed? This is very important in the study and can be mentioned in title itself.

The manuscript does not include information about any underlying condition in the infants which might be affecting the RSV infection irrespective to palivizumab prophylaxis.

I understand the limitation of the study. But because palivizumab has very short half-life, it is very important to know the number of doses infants received and when they receive the dose/s.

Unfortunately, many infants did not receive the dose despite of the guideline, those infants should be followed up very carefully for any symptoms/illness.

Preterm infants with gestational age of ≤32 weeks who will be less than 6 months old in the winter season are eligible to receive palivizumab doses under insurance coverage. This study shows similar RSV infections for both very preterm (<32 weeks) and preterm (>32-37 weeks) infants in non-prophylactic groups. Do the authors think that they should collect and analyze more data so that the policy can be extended for preterm (>32-37 weeks) infants?

Reviewer 2 Report

Comments and Suggestions for Authors

Thank you for the opportunity of reviewing this manuscript. The authors demonstrated the real-world efficacy of palivizumab in reducing the incidence of serologically-confirmed RSV infection in pre-term neonates. I think the methods, results, and discussion are presented nicely. The main limitations of the manuscript are stated clearly. I only have a very minor suggestion which may help to improve the readability of the manuscript:

1. In Figure 2, the authors present a plot of antibody concentrations within several patient groups. Asterisks (*) are used in the figure which correlate with the degree of statistical significance, but it would be helpful if the corresponding p-value ranges are provided as well (e.g. **** = p<0.0001)

I otherwise have nothing to add. Well done to the authors!

Round 2

Reviewer 1 Report

Comments and Suggestions for Authors

Authors have responded to all the questions well and improved the manuscript significantly.